# A Compact Design of Underwater Mining Vehicle for the Cobalt-Rich Crust with General Support Vessel Part A: Prototype and Tests

Chao Xie [1,2,*], Lan Wang [1], Ning Yang [2,*], Casey Agee [2], Ming Chen [2], Jinrong Zheng [2], Jun Liu [2], Yuxiang Chen [2], Lixin Xu [3], Zhiguo Qu [2], Shaoming Yao [1], Liquan Wang [1] and Zongheng Chen [4]

1    School of Mechanical and Electrical Engineering, Harbin Engineering University, 145 Nantong Avenue, Harbin 150001, China; wanglan@hrbeu.edu.cn (L.W.); yao.sm@hotmail.com (S.Y.); wangliquan@hrbeu.edu.cn (L.W.)
2    Department of Deep-Sea Technology and Engineering, Institute of Deep-Sea Science and Engineering, Chinese Academy of Sciences, 28 Luhuitou Road, Sanya 572000, China; Casey@idsse.ac.cn (C.A.); Chenm@idsse.ac.cn (M.C.); Zhengjr@idsse.ac.cn (J.Z.); liu-jun@idsse.ac.cn (J.L.); chenyux@idsse.ac.cn (Y.C.); quzhiguo@idsse.ac.cn (Z.Q.)
3    China Merchants Marine and Offshore Research Institute Co., Ltd., Shenzhen 518067, China; xulixin@cmhk.com
4    Guangzhou Marine Geological Survey, No. 188 Guanghai Road, Huangpu District, Guangzhou 510760, China; czhgs@foxmail.com
*    Correspondence: xiechao@idsse.ac.cn (C.X.); yangning@idsse.ac.cn (N.Y.)

**Abstract:** This paper proposed a compact design of the subsea cobalt-rich crust mining vehicle with a general purpose support vessel for subsea resource exploration, sample collection, and research. The necessary functions were considered in the concept design, including walk, crushing/mining, sample collection, cutter head adaptation, vehicle orientation, crust texture measurement, awareness, positioning, and navigation. The prototype was tested in both tank and subsea environment. The sea trials were carried out with the support of a general purpose support vessel. The track design worked well in both the tank and subsea environment and the mining vehicle walked smoothly in the sea trial. The crust was crushed to the size of 2 mm and 10 mm with different cutting parameters and successfully collected by the jet pump, 6 kg in total. The crust texture was measured by the onboard sonar successfully and can be used for cutting parameter selection. The cameras captured the images of the subsea environment, but the actions of crushing and sample collection produced plumes, which blocked the camera vision. In the situation, the front image sonar can be used to keep the vehicle away from big rocks. The mining vehicle is not limited to the mining and sampling of subsea cobalt-rich crust. Most of the subsea solid resources on the seabed can be considered to use the compact mining vehicle for sampling and related research. The only issues to be considered are the crushing ability and sample size required.

**Keywords:** ocean mining; mining vehicle; sea trial; tank test; cobalt-rich crust

## 1. Introduction

The oceans cover about 71% of the Earth's surface, and much more mineral resources are available on the sea floor than on land, including nickel, copper, and manganese. The interest in deep-sea mining has grown substantially in the last decade [1] for sea floor massive sulfides (SMSs) near hydrothermal vents, poly-metallic nodules on the abyssal sea floor, and cobalt-rich crusts on sea mounts. Deep-buried mineral deposits provide options for supplying metals, which is essential for the electronic and evolving green technology industries, such as electric vehicles (EVs), solar panels, etc. However, subsea commercial mining is a significant challenge due to the high cost and uncertain environmental impacts.

The midwater (from 200 m to 5000 m) ecosystems occupy over 90% of the biosphere, contain fish biomass 100 times greater than the global annual fish catch, connect shallow and deep-sea ecosystems, and play critical roles in nutrient regeneration, and provisioning of harvestable fish stocks [1]. Deep-sea mining generates noise and plumes, which are pollutions to marine ecosystems as well as humans.

DeepGreen Metals owns three companies: Tonga Offshore Mining Limited (TOML), Nauru Ocean Resources, Inc. (NORI), Nauru, and Marawa Research and Exploration Ltd., Tarawa, Kiribati, with three exploration contract areas. A deep-sea database was created for the Clarion-Clipperton Zone (CCZ) in the eastern Pacific Ocean. Three webinars were organized in May and June 2020 and a white paper published [2]. which presented the work plan for an in-depth life cycle assessment study that compared the impacts of two sources of metals, i.e., land ores and deep-sea poly-metallic ores. The study focused on four metals used in EV battery cathodes and wiring: nickel, cobalt, manganese, and copper. Deep-sea mining can be a desirable alternative in all aspects compared with terrestrial mining [3]. However, hazard identification and risk assessment are needed before the development of yet nonexistent (or unproven) technologies [4].

The cobalt-rich crusts and SMSs occur in similar circumstances: both need to be split from the sea floor or rocks and then collected by specific machines or devices and transported from the subsea to a vessel on the sea surface by a lift system. Thus, their mining techniques would be similar as well.

Halkyard proposed a likely mining machine for the cobalt-rich crust that consists of a four-crawler device, cutting drums, slurry pump, and hydraulic pipe lift system in 1985 [5]; however, no sea tests were conducted to verify its performance.

In 2000, Nautilus Minerals first created a work plan to explore SMSs and hired Soil Machine Dynamics Ltd. (SMD), Newcastle, UK, to develop commercial mining machines in 2007. In the early 2010s, an exploration permit and exploitation lease were secured from the government of Papua New Guinea. With an expectation for production in 2019, they developed a mining machine and carried out a cutter underwater test in 2017. The mining machine consisted of two crawlers and a cutting drum, a centrifugal slurry pump was used for collection. However, it bankrupted in 2019 due to opposition from the residents and financial setbacks [6]. Nevertheless, it was successfully restructured by the Deep Sea Mining Finance Limited (DSMF) in 2019. DSMF held the project of Solwara 1, three subsea production tools, a large-scale mining system, and at least 12 patents for various deep-sea mining system technical components, including drilling, subsea mining, and riser and hoisting systems.

Japan Oil, Gas and Metals National Corporation (JOGMEC) launched a feasibility research project for SMS mining in 2008 and developed a test machine in March of 2012. The machine was tested in an offshore environment to excavate SMSs from the sea floor in the Okinawa waters in November 2012, where it successfully collected SMS samples. The machine was a 184 kW-class system weighing 20 metric tons with four suspension crawlers, cutting drum, and centrifugal slurry pump for collection. It collected approximately 25 kg of SMS samples placed in ore storage [7]. During the mining tests, two ships were adopted for the ore collecting test and ore lifting support [8].

JOGMEC built a submersible pump system and mining test machine and carried out a cutting, dredging, walking (on land), and lifting test. In July 2020, JOGMEC collected 649 kg of cobalt and nickel-rich seabed crust [9].

The Viable Alternative Mine Operating System (VAMOS) project was funded by the European Union's Horizon 2020 program from 2015 to 2018. The project prototyped a mining system with the mining machine of SMD to extract raw materials from inland waters and successfully demonstrated and tested the system in Silvermines, Ireland in October 2018. Track tests were performed, but the other tests were conducted in stationary conditions underwater, e.g., cutter movement, auger collection, and backhoe operation [10]. The raw sensor data, terrain maps, positioning and navigation data, and system information were transferred to a human-machine interface (HMI) via a local area network. HMI was

based on a custom-built virtual reality (VR) application and built on top of a Unity gaming engine. It provided a 3D VR model of all mining operations [11,12]. VAMOS provided a new mining technique that was expected to be used for reopening abandoned and flooded open-pit mines [13] and can also be applied in marine mining.

The current cobalt-rich crust mining vehicles were too heavy for the most of general purpose ROVs. Therefore, the dedicated vehicles were required for support, which was not suitable for mineral sample collection and resource exploration.

Therefore, this paper proposes a compact cobalt-rich crust mining vehicle which can be operated with the support of a general support vessel, which consists of two crawlers, cutting drum, and water jet pump. It covers most of the essential mining functions and can adopt the existing ROV and recovery system for support, which can significantly save the cost of sea trial. Firstly, the requirement of mining vehicles was analyzed; the mining vehicle concept was created and prototyped with regard of the functions of walk, crushing, adaptation, collection, orientation, awareness, and position and navigation, the working modes are defined and analyzed in Section 2. Secondly, the mining testing system is introduced for the sea trial in Section 3, as well as the test procedure. At last, the tank experiments and sea trials were carried out.

## 2. Function and Design of the Mining Vehicle

Cobalt-rich crusts deposits exist on slopes, summits, and platforms of seamounts and guyot structures in the oceans, the classical cone-shaped seamounts (slop angle from 12° to 16°) and the guyots are the most typical landforms for the mining of the cobalt-rich crust [14,15] in the water depth range of 450~7000 m. Moreover, they distribute in the water depth range of 450~7000 m. The main distribution water depth of the Pacific Ocean is 1000~3500 m, the main distribution water depth of the Atlantic Ocean is 2000~4000 m, and the main distribution water depth of the Indian Ocean is 1500~5500 m. If the crust thickness delineation is 2 cm, the real thickness is 2.5~5.2 cm. If the delineation is 4 cm, the real thickness is 5.5~7.1 cm [16]. The substrate of the crust includes breccia, basalt, phosphorite, limestone, transparent clastic rock, and mudstone. The adhesion between cobalt-rich crust and substrate is strong or weak. The key element to the mining is to collect crusts as efficiently as possible and reduce the possible dilution, such as substrates collected [17].

Generally, the slope angle of the working place, the thickness of cobalt-rich crusts, the mechanical strength of crust and substrates, and the variation in microtopography need to be considered during the design.

### 2.1. Requirements

The requirements of the design of the subsea vehicle include general requirements, essential requirements, and additional requirements.

(1) General requirements:

- Meeting the demands of the mining test with the minimum cost and a good maintainability to support broader marine mining research;
- The general support vessel can be used, particularly the launch and recovery system and umbilical.

(2) Essential requirements:

- The mining vehicle can work on the hard seafloor of the slope and adapt to the possible micro-topography;
- The crushing tool can peel off the cobalt-rich crust covered on the rock and crush it to 15–30 mm size;
- The cutter head needs to move up and down to follow the microtopography;
- An efficient and reliable particles collection is required. collected particle size is 0–100 mm;
- Keeping the mining vehicle at the same heading with the support vessel during the launch process.

(3) Additional requirements:

- The crust thickness measurement determines the depth of cutting, which is an important factor for the ore dilution rate control. The ideal way to measure the crust thickness by a sub-bottom profiler is contactless measurement;
- Positioning and navigation are required as the complex terrain and dark environment.

### 2.2. Functions

According to the design requirements, the mining vehicle should have the functions of walk, crushing, adaptation, and collection. As shown in Figure 1, the vehicle adopts the crawler chassis to walk, cutter head to peel the crust from the bedrock, and hydraulic cylinder arm to control the cutter head adaptive to microtopography and the hydraulic collection device behind the cutter head to collect the fragmented crust. Table 1 shows the essential parameters of the mining vehicle.

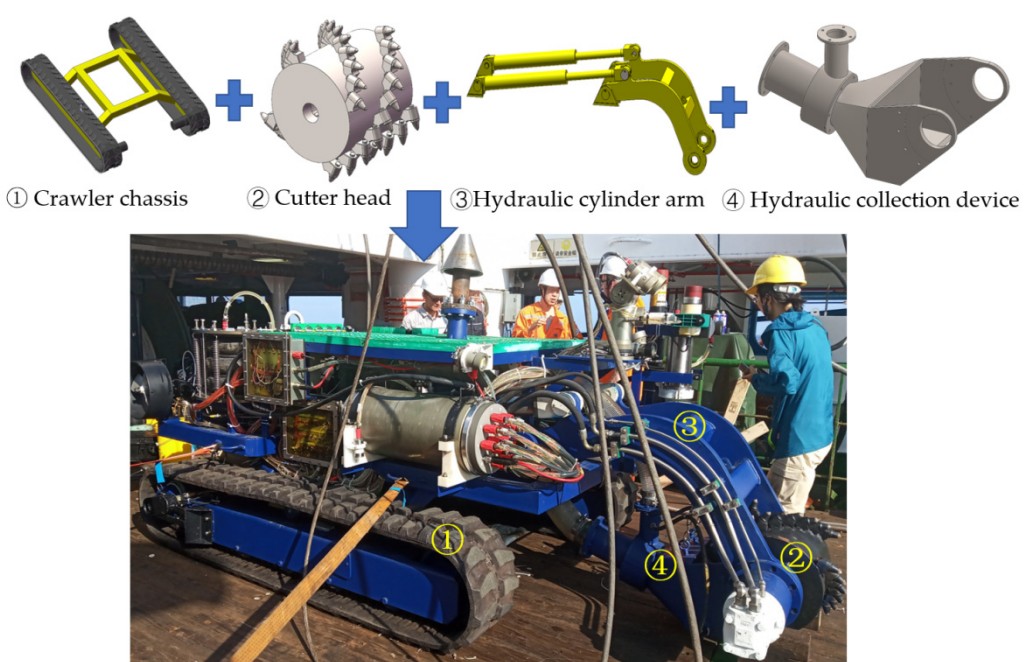

**Figure 1.** Essential functions of the mining vehicle.

**Table 1.** Essential parameters of the mining vehicle.

| Parameters | Unit | Value |
| --- | --- | --- |
| Weight | ton | 4.1 (Air)/3.0 (Water) |
| Size | m | 4.3 (L) × 2.3 (W) × 1.6 (H) |
| Track gauge | m | 1.65 |
| Maximum walking power | kW | 25 |
| Maximum crushing power | kW | 22 |
| Collection power | kW | 18.6 |

#### 2.2.1. Walking

Crawler chassis is widely used in engineering and agricultural equipment. For the sea floor miner, tracked vehicles are preferred compared with wheeled or legged ones due to the large contact area with the ground providing better floatation and larger traction forces [18]. Two typical tracks are available: steel tracks and rubber tracks.

A steel track consists of track links and track shoes, and the track links connect each other by track pins and track shoes mounted under the track links [19], as shown in Figure 2a. It is widely used in the products such as crawler excavators, crawler loaders,

bulldozers, etc., of construction machinery companies such as XCMG Group, Sany Heavy Industry, Caterpillar, Komatsu, etc. The track teeth insert into the ground to provide the driving force required for vehicle body walking. The steel track is much heavier than rubber tracks, thereby the traction enhanced as well as the stability and floatation, especially on the loose ground. The damaged parts can be simply replaced, which makes the maintenance cost efficient, and the track shoes can be modified for different application scenarios. For example, a rubber block or metal plate integral vulcanized rubber can be added to the track shoe to prevent the track shoes from damaging the concrete ground. The steel track is also applicable in marine engineering, such as mineral extraction vehicles, trenchers, etc. Table 2 shows the comparison between the two types of tracks.

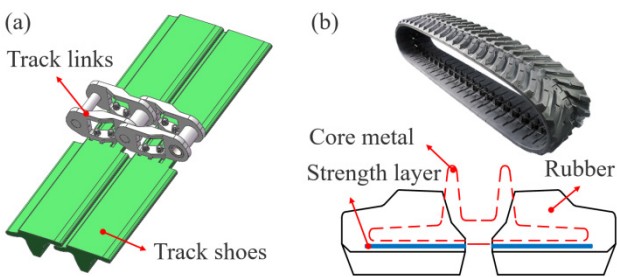

**Figure 2.** Structure of tracks: (**a**) steel tracks and (**b**) rubber tracks (Yachoo).

**Table 2.** Comparison between the two types of tracks.

| Type | Products and Application | Advantages and Disadvantages |
|------|--------------------------|------------------------------|
| Steel tracks | **Land application:**<br>Company: XCMG, Sany, Caterpillar, Komatsu, etc. including Mining machines, Excavators, Crawler cranes, Bulldozers, etc.<br>**Marine application:**<br>SMD Trencher, Mineral extractors, and deep-sea mineral extraction vehicle, Royal IHC "I-TRENCHER" Trencher, Seatools "PATANIA II" Mining vehicle and Trencher, etc. | **Advantages:**<br>Best traction;<br>Easily removable;<br>Rarely damaged.<br>**Disadvantages:**<br>More expensive;<br>Large vibration;<br>More weight;<br>lower speeds;<br>Poor corrosion resistance. |
| Rubber tracks | **Land application:**<br>China YTO Crawler-type tractors and Caterpillar harvesters,<br>KUBOTA Caterpillar harvesters, John Deere Caterpillar harvesters, etc.<br>**Marine application:**<br>Germany GEOMAR "VIATOR" ROV, Japan JAMSTEC "ABISMO" ROV, Germany Jacobs University "Wally" and "iWally", American MBRI "Benthic Rover", Germany AWI "TRAMPER", etc. | **Advantages:**<br>Extremely affordable;<br>Less weight;<br>Fewer vibrations;<br>Work well in wet and muddy conditions.<br>**Disadvantages:**<br>Not great with durability. |

A rubber track has a simpler structure, as shown in Figure 2b. It consists of a core metal, strength layer, and rubber [20]. The core metal and the strength layer provide the support and traction for the vehicle instead of the track link of the metal track, so it is much lighter and more affordable than steel tracks, and it is nearly maintenance free as it is a one-piece crawler. However, the whole track after wear needs to be replaced. It is mainly used for products of agricultural crawler machinery and construction machinery companies, such as China YTO, KUBOTA, and John Deere. In addition, there are rubber tracks without core iron, which are lighter and are mainly used in wheeled vehicles and

tracked robots, such as JAMSTEC "ABISMO" ROV, Jacobs University "Wally" and "iWally", MBRI "Benthic Rover", etc., [21]. The comparison is listed in Table 1.

Generally, the working time of offshore test equipment is much shorter than the construction machinery. The rubber track with corn metal is used for the mining vehicle to make it light and cheap and ensure good traction and support as well. However, steel tracks are still the best choice in the cases of long-term operation. Table 3 shows the crawler chassis specification.

**Table 3.** Crawler chassis specification.

| Parameters | Unit | Value/Description |
|---|---|---|
| Manufactory | - | China Kemer |
| Track type | - | KRT3500 |
| Track width | mm | 350 |
| Working pressure | MPa | 19 |
| Weight | kg | 600 |
| Track gauge | m | 1.65 |
| Wheel diameter | mm | 300 |

### 2.2.2. Crushing

Three cutting strategies are available, including down-milling, up-milling, and slotting [22] as shown in Figure 3a–c. In the up-milling the cutting velocity keeps the same direction with the feed and in the down-milling it is opposite to the feed direction [18].

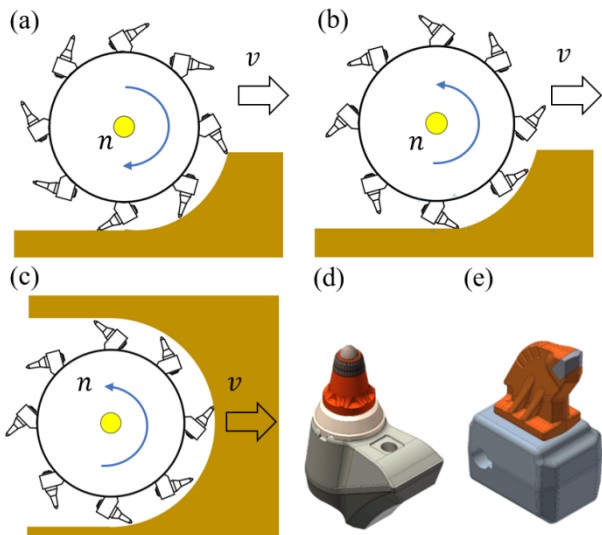

**Figure 3.** Cutting modes and cutter bits types: (**a**) up-milling mode, (**b**) climb-milling mode, (**c**) slot-milling mode (lateral view of the cutter head) (**d**) conical cutters (BETEK) (**e**) radial cutters (BETK).

In the up-milling, the fragmented particles come out from the front of the cutter head and are left on the uncut surface. If the particles are not collected in time, the extra power will be consumed. Up-milling was extensively adopted in pavement milling machines. In the down-mining, the fragmented particles come out behind the cutter head. Down-milling was adopted in the SMD's "Bulk cutter" and mining vehicle of the "VAMOS" project. In the slotting mode, the big contact angle makes the particles excessively crushed. Slotting mode was extensively used in longwall shearers. The comparison is listed in Table 4.

**Table 4.** Three cutting modes.

| Cutting Modes | Applications | Possible Problems |
|---|---|---|
| Up-milling | Pavement milling machine | Fragmented particles place on the uncut ground may cause excessive crushing; Additional walking resistance from the cutter head. |
| Down-milling | Continuous miners; SMD's "Bulk cutter"; VAMOS 's Mining vehicle | More vibration. |
| Slot-milling | Long wall shearers | Fragmented particles cannot leave cutter head in time will cause excessive crushing; Additional walking resistance from the cutter head. |

Two typical cutters are available, conical cutter and radial cutter, as shown in Figure 3d,e. Radial cutters are suitable for excavating soft materials and are more efficient than conical cutters. The service life of the conical cutter is better due to the conical profile [22]. The conical cutters can be welded spirally on the drum of the cutter head [23].

The conical cutter was selected in this mining vehicle design and spirally arranged on the drum of the cutter head. The cutter head is driven by a hydraulic motor, which is controlled by a proportional valve, and the speed can be set during the operation. Table 5 shows the cutter head specification.

**Table 5.** Cutter head specification.

| Parameters | Unit | Value/Description |
|---|---|---|
| Cutter manufactory | - | BETEK |
| Type | - | BC68 |
| Tips angle | ° | 75 |
| Cutter length | mm | 100 |
| Size of cutter head | mm | Diameter $\phi$ 550 × Width 400 |
| Installation angle of cutter | ° | 45 |
| Number of spiral lines | - | 2 |
| Torque | N·m | 1310 |
| Rated speed | rpm | 125 |
| Max power | kW | 22 |

### 2.2.3. Adaptation

Due to the crusts, nodules, sediments, etc., the floor surface fluctuates from a few centimeters to several meters [24]. The thickness of the crust needs to be detected, such as by acoustics devices, and then adjust the height of the cutter head by the hydraulic cylinder to make the cutter head adaptive to the floor surface [25]. However, the acoustic detection technology is not so reliable currently and more tests are needed [26]. The acoustic detector was installed and tests were carried out.

In the current research, utilizing the mechanical property difference between the crust and substrate, the cutting force based self-adaptive control is proposed and tested as well. The adaptation system is as shown in Figure 4a, it consists of the cutter head, hydraulic cylinders, relief valve, and oil supply. Hydraulic cylinder parameters are shown in Table 6.

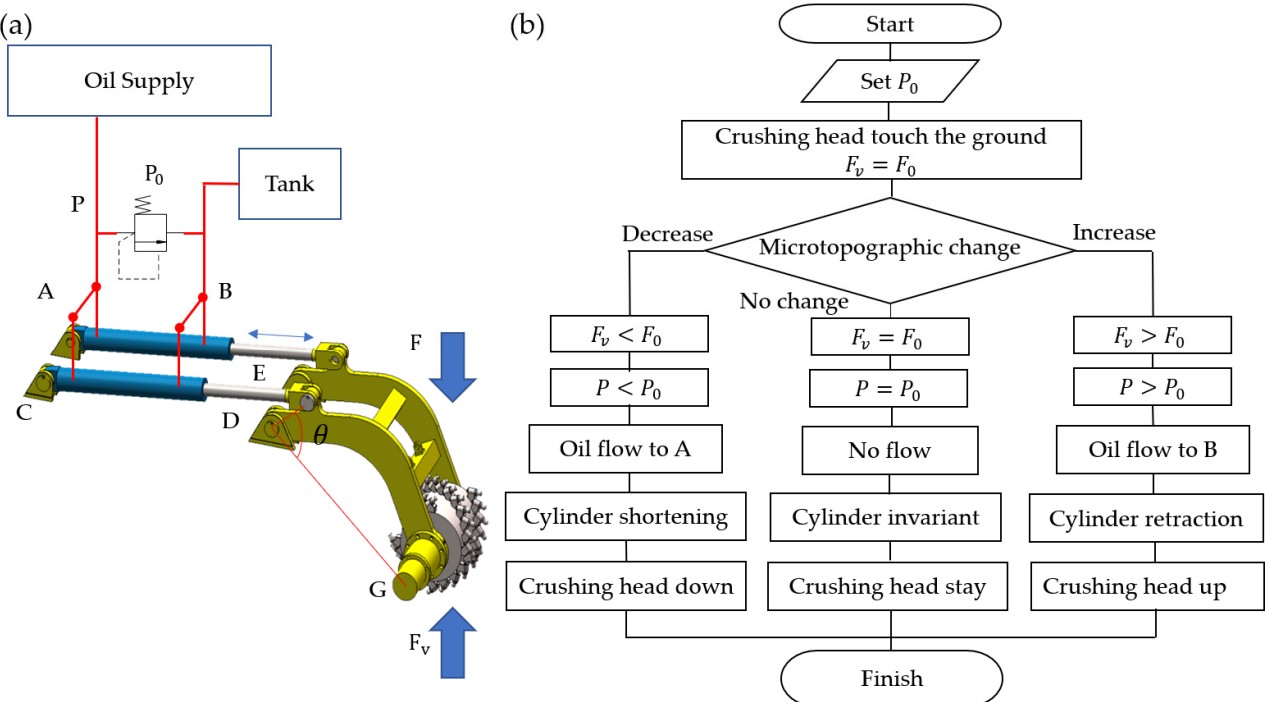

**Figure 4.** Floating function: (**a**) working principle and (**b**) working flow chart.

**Table 6.** Hydraulic cylinder parameters.

| Parameters | Unit | Value/Description |
|---|---|---|
| Manufactory | - | Shanxi Gaohang Hydraulic inc. Jinzhong, China. |
| Size | mm | $\varphi 100 \times \varphi 50$ (D/d) $\times$ 1750 (L) |
| Stroke | mm | 200 |
| Supply pressure | MPa | 0–19 |
| Cylinder distance | mm | 560 |
| Hinge distance | mm | CD = 1500; DE = 300; DG = 1172 |
| Head posture angle | ° | $\theta = 69.6$ |

The control flow chart is as shown in Figure 4b. If the cutting force is lower than the setting value, the cutter head must be lowered to increase cutting depth; if the cutting for is higher than the setting value, the cutter head must be lifted to reduce the cutting depth.

### 2.2.4. Collecting

Centrifugal slurry pumps have been utilized in underwater engineering [27] which is not suitable for the mining vehicle due to the size and operation depth. A jet pump is another option, which transfers energy from a high-velocity jet to a low-velocity stream. Jet pumps are widely used marine projects because of the simplicity and high reliability [28].

In the current work, an annular jet pump is used for the crust collection, as shown in Figure 5.

Firstly, the crushing head breaks the crusts into particles and the particles go into the settlement chamber and are sucked into the jet pump. The particles are collected and stored in a rock case, the primary flow is supplied by slurry pumps, and it is driven by a 3000 v electric motor. Table 7 shows the main hydraulic collection parameters.

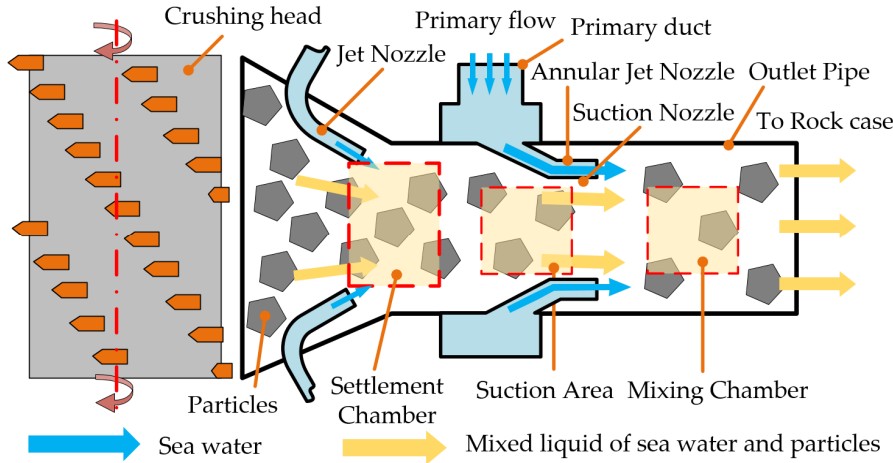

**Figure 5.** Hydraulic collection schematic diagram.

**Table 7.** Hydraulic collection parameters.

| Parameters | Unit | Value/Description |
|---|---|---|
| Annular nozzle diameter | mm | $\phi$130 |
| Outlet pipe diameter | mm | $\phi$150 |
| Primary flow speed | m/s | 5.5 |
| Slurry pump manufactory | - | Zhejiang Yangtze River Pump Co., Ltd., Wenling, China. |
| Slurry pump type | - | 100FSB-40L |
| Flow | m³/h | 100 |
| Lift | m | 40 |
| Electric motor manufactory | — | Tianjin Premier ESP Pumping System Co., Ltd., Tianjin, China. |
| Voltage | VAC | 3000 |
| Phase | - | 3 |
| Rated speed | rpm | 2900 |

### 2.2.5. Orientation

Two hydraulic thrusters (propellers) are installed in front and behind the vehicle to produce side forces for direction control, particularly in the vehicle launch and recovery, as shown in Figure 6, and Table 8 shows the main parameters of the propeller.

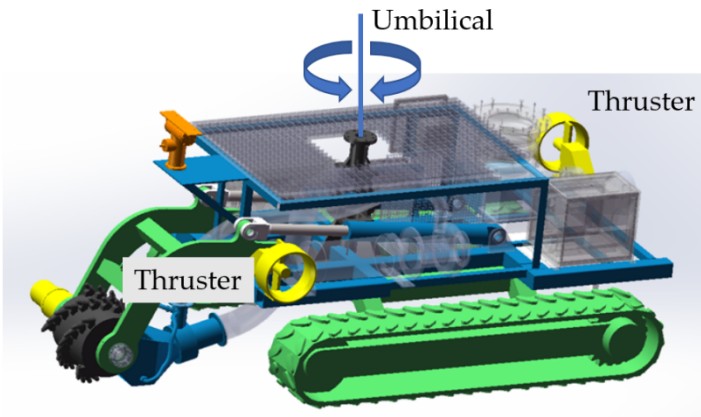

**Figure 6.** Schematic diagram of orientation function.

**Table 8.** Propeller parameters.

| Parameters | Unit | Value/Description |
|---|---|---|
| Supplier | - | Shanghai Jiaotong University, Shanghai, China |
| propeller diameter | mm | 300 |
| Speed | rpm | 800 |
| Thrust force | kgf | 289 (@250bar) |

### 2.3. Additional Functions

The additional functions include crust measurement, awareness, positioning, and navigation.

### 2.3.1. Crust Bedding Structure Measuring Device

A high-frequency submersible sub-bottom profiler (HFSSP) is employed for the measurement of the thickness of the cobalt-rich crusts in this sea trial. The crust thickness results can be used for adjusting the cutting depth and reduce the substrate cutting.

It was mounted in the front of the vehicle and its specification is listed in Table 9.

**Table 9.** HFSSP specifications.

| Parameter | Unit | Value |
|---|---|---|
| Central frequency | kHz | 110 |
| Bandwidth | kHz | 95–125 |
| Ping rate | Hz | 8 |
| Probe depth | m | 2–5 |
| Depth resolution | cm | 2.5 |
| Operation height | m | 1–5 |
| Working depth | m | 7000 |

### 2.3.2. Awareness System and Positioning and Navigation System

The left of Figure 7 shows the awareness system, including a TV system, image sonar, attitude indicator and depthometer. A TV system provides real-time videos and images for the operator. Image sonar is the necessary equipment to ensure the safety of the mining vehicle during the launch and walking, and it is used to detect the terrain outside the field of the TV system. Attitude indicator and depthometer feedback the altitude and depth to positioning and navigation system, which control the direction and trajectory of mining vehicle.

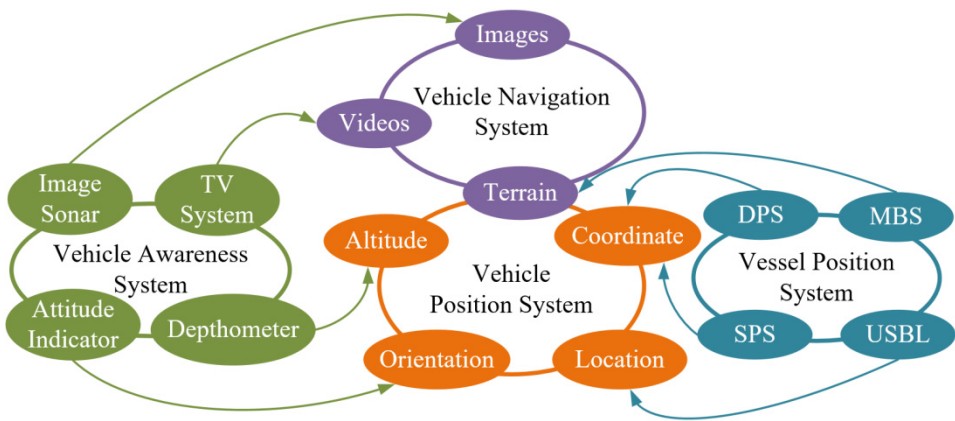

**Figure 7.** Awareness system and Positioning and navigation system.

The right side of Figure 7 shows the vessel positioning system, including the Dynamic Positioning System (DPS), Multi-beam Bathymetry System (MBS), Ship Positioning System

(SPS), and Ultra-Short Baseline (USBL). DPS and SPS can determine the vessel position in selected coordinates, while USBL can determine the vehicle's position relative to the support vessel. MBS is used to obtain the topographic map of the operation area.

The middle of Figure 7 shows the positioning and navigation system working principles. These functions are realized by a vehicle awareness system and vessel positioning system.

### 2.4. Specifications of the Mining Vehicle

Figure 8 shows the entire system composition of the vehicle. Generally, it includes eight parts: crawler chassis, cutter head, orientation module, hydraulic collection, electrical system, awareness system, hydraulic system, and positioning and navigation system. The main functions and modules are also shown in the same figure. HFSSP is not shown here as it is a separate system. The electrical system provides energy and communication for the whole system.

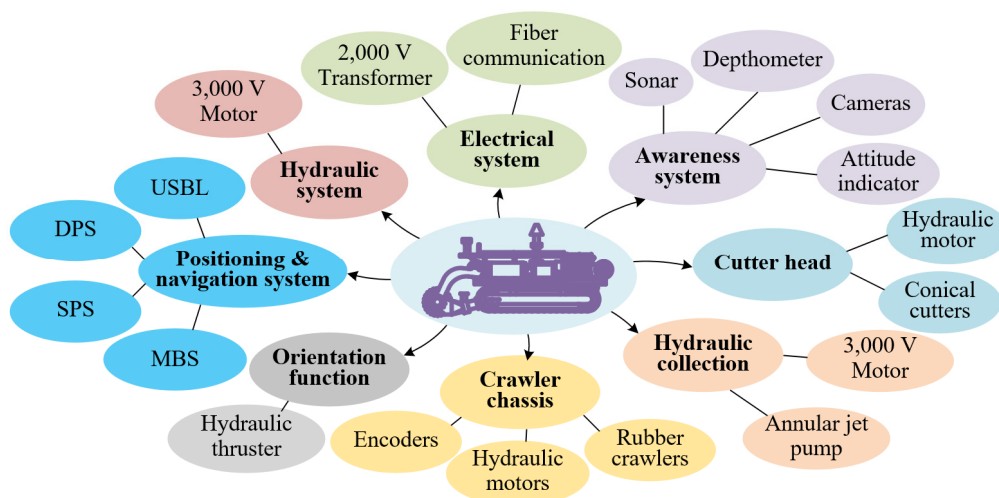

**Figure 8.** Vehicle components.

### 2.4.1. Parameters

The vehicle parameters are listed in Table 10. It is nearly three metric tons in water, and the operation depth is up to 4500 m. All the components passed a 60 MPa pressure test. The supply voltage is 3000 V, and the power is 70 kW. The vehicle specifications are shown in Table 10.

**Table 10.** Vehicle specifications.

| Parameters | Description/Value |
| --- | --- |
| Working depth | 4500 m |
| Weight | 4.10 T (air)/3.00 T (water) |
| Size | 4.3 m (L) × 2.3 m (W) × 1.6 m (H) |
| Power | 70.6 kW |
| Voltage | 3.000 V |
| Ground pressure | 20.8 kPa |
| Hydraulic system pressure | 19 MPa |
| Walking speed | 0–0.05 m/s |
| Speed of cutter head | 0–120 rpm |
| Floating range | −125~250 mm |
| Power mode | Umbilical |
| Power configuration | Hydraulic motor: 50.0 kW, 3.0 kV<br>Water pump motor: 18.6 kW, 3.0 kV<br>Control system transformer: 2.0 kW, 2.0 kV |
| Recovery system | A-Frame, Umbilical winch, Recovery equipment |
| Camera | HD camera × 4 (1080p),<br>PAL camera × 2 (650 PVL),<br>Pan and tilt × 1 |
| Onboard equipment | Image sonar, depth sensor, sub-bottom profiler |

### 2.4.2. Working Modes

There are two working modes: seafloor mode and seawater mode.

(1) Seafloor mode

Seafloor mode includes walk, cutting, adaptation, and collection functions. There are cliffs, deep ditches, boulders, etc., on a seamount and the vehicle operation environment is much more complicated than that of ROV. In order to ensure the safety of the mining vehicle and support vessel, the mining vehicle is manually operated on the support vessel.

(2) Seawater mode

Seawater working mode is for the launch and recovery processes. The operator sets a direction angle in the operation interface and the thrusters drive the vehicle to the right angle automatically.

## 3. Mining Test System and Procedure

The test of the mining vehicle needs the support vessel and umbilical system. The entire system for the test is as shown in Figure 9.

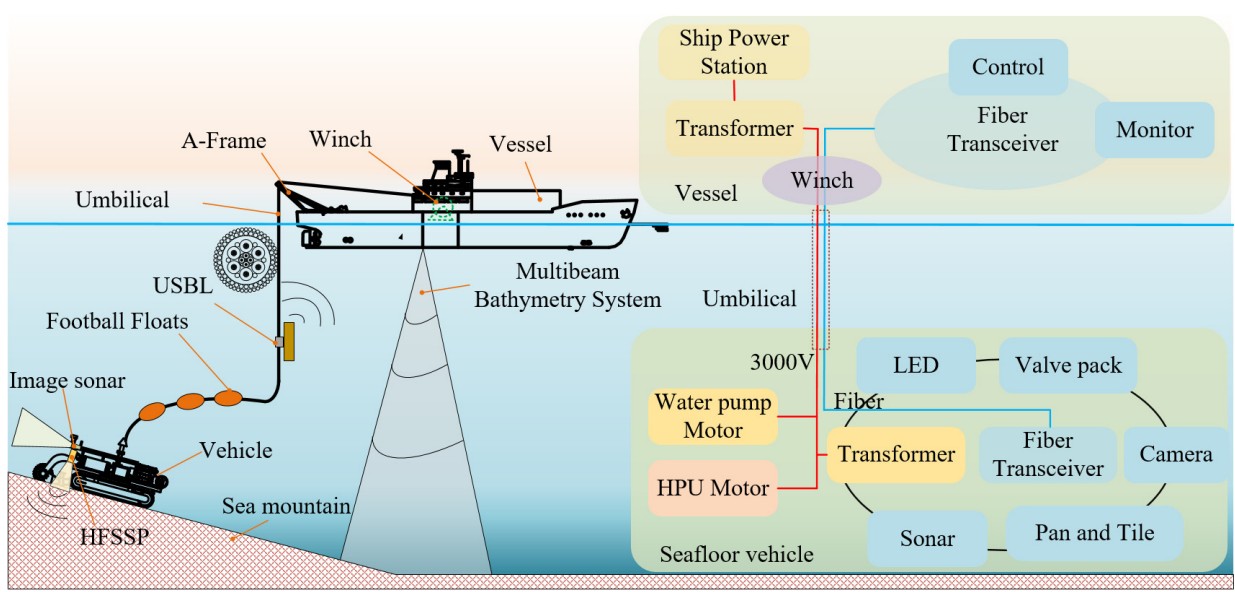

**Figure 9.** Left: schematic diagram of mining test system; Right: structure of the electrical system.

### 3.1. Mining Testing System

Figure 9 left shows that the mining test system includes the vehicle, umbilical, and support vessel above water. The vehicle works on the seafloor, and the vessel works on the sea surface, connected by the umbilical. Football Floats were installed on the umbilical to keep it slack near the vehicle and allow it to move with a small force. The ultra-short baseline (USBL) was mounted on the umbilical below the vessel to indicate the vehicle's position to the vessel. A multi-beam bathymetry system (MBS) can scan submarine topography for the vehicle.

The vessel supplies 3000 V of power to the vehicle. It drives a water pump motor and HPU motor directly and supplies power through the transformer to the electric system. The vessel communicates with the vehicle through the optical fibers in the umbilical, which can realize real-time monitoring and control.

Figure 10 shows the support vessel, Hai Yang Liu Hao, which is a scientific investigation ship of Guangzhou Marine Geological Survey (GMGS). Its size is 106 m (L) × 17.4 m (W) × 8.3 m (H), and the full load displacement is 4600 t. It is equipped with a dynamic positioning system (DPS), a ship positioning system (SPS), a multi-beam bathymetry system (MBS), USBL underwater acoustic positioning system, A-frame, winch, and other systems. The winch, A-frame, and DPS are used for launch and recovery.

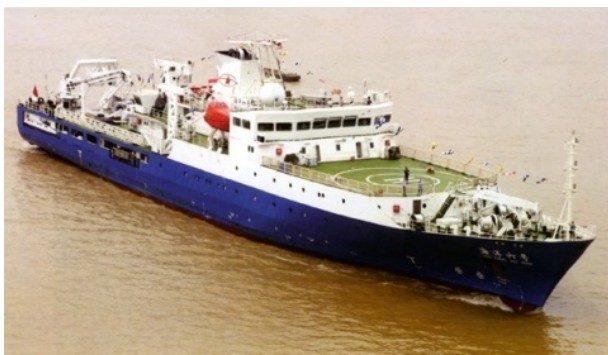

**Figure 10.** Support vessel "Hai Yang Liu Hao".

Figure 11 shows the vehicle is launched the launch and recovery system of the support vessel.

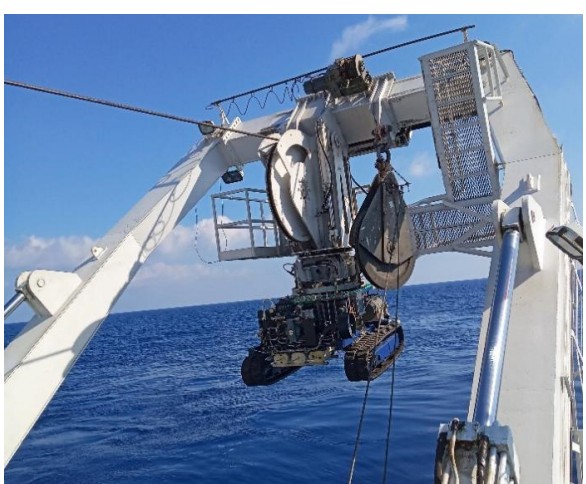

**Figure 11.** Launch and recovery system of the mining vehicle.

The umbilical connects the vessel and the vehicle and contains six power cables and four fiber cables inside. The rated voltage is 3300 V, and the power is 70 kW. The outer diameter is 21.21 mm, the paid load is 56.6 kN, and the maximum operation depth is 6000 m.

### 3.2. Test Procedure

The mining test procedure is as follows:

Step 1: Select test location. The preferred terrain is a seamount with a gentle slope no more than 25°.

Step 2: Terrain confirmation before test. After reaching the selected operation area, MSB needs to scan the operation area again to obtain a detailed multi-beam topographic map and select the operation point again.

Step 3: Support vessel positioning. Turn on DPS of the support vessel to the selected coordinates before launch.

Step 4: Launch the mining vehicle and turn on the orientation function.

Step 5: Landing area safety confirmation. Find the flat ground by TV system for landing.

Step 6: Turn off the orientation function, change the mining vehicle control to manual control, and finish the walking and crushing test.

Step 7: Observe the environment around the mining vehicle to make sure there are no rock walls.

Step 8: Slowly recover and tighten the umbilical, observe the deviation direction of the umbilical of the mining truck through the camera, and keep the umbilical in a vertical state. If not, move the support vessel to the right position.

Step 9: Recovery the umbilical quickly and keep the mining vehicle a certain height above the ground.

Step 10: Turn on the orientation function, recovery the mining vehicle.

Step 11: One walking and crushing test is finished.

## 4. Tank Experiments

Tank experiments are required to test functions such as suction, cutting, and walk before the sea trial.

### 4.1. Walking Function Test

The vehicle underwater walking test is on the floor of a 22 m (L) × 10 m (W) × 20 m (D) freshwater tank. The umbilical is hung up by a crane. The walking and turning are tested by manual control. Figure 12 shows the walking trajectory of the vehicle, captured by the camera on the vehicle.

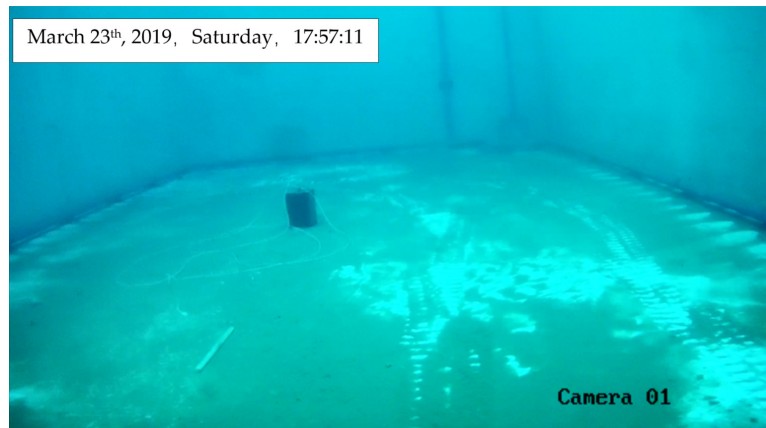

**Figure 12.** Walking trajectory of tank test.

### 4.2. Collection Function Test

Figure 13 shows the collection function test method and results. A hose connects directly to the bucket of the collection device, the water pump was turned on and the particles were placed into the bucket through the hose. The particles were sucked into the annular jet pump and the collection bag filters out the particles at the outlet of the jet pump. The biggest particle was 76 mm in length. The collection function is even better with the assist of the rotation of the crushing head.

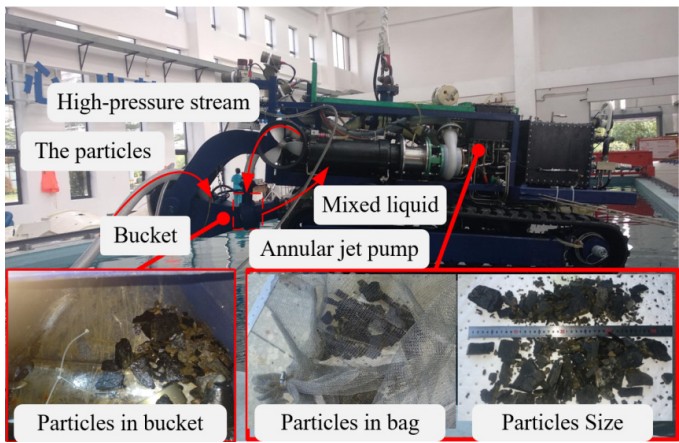

**Figure 13.** Collection function test method and results.

### 4.3. Cutting Function Test

Figure 14a shows the tank test of the vehicle, the sample block is fabricated of cement and the strength is 60 MPa measured by a rebound apparatus. The blocks were placed at the bottom of the tank, the vehicle was driven on the surface and the sample surface was cut for 1 m in length. The cutting produced a groove on the sample surface is 34 mm in depth and 400 mm in width, as shown in Figure 14a,c. The cutting test in the tank was successful.

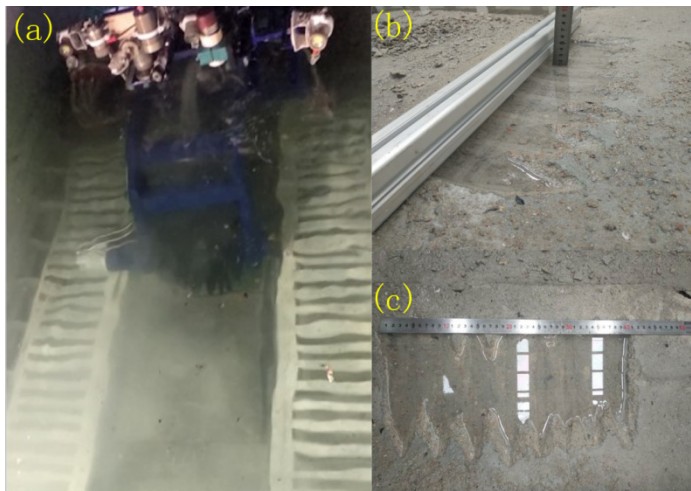

**Figure 14.** (**a**)Tank testing, (**b**) depth of the grooves, (**c**) grooves on the cement block.

## 5. Sea Trial

The sea trial was conducted the South China Sea. The objective was to test the vehicle's functions and the launch and recovery process. A seamount with a hard floor is available with a depth of 2500 m and suitable for the test.

Seven sea trials were conducted in total, two trials were in depth less than 1000 m, and five were in depth from 1000 to 2500 m. The details are listed in Table 11. The system of HFSBP was tested seven times, and it worked correctly in five of them.

**Table 11.** Overview of the launches.

| Dive | Location | Depth | Results |
|------|----------|-------|---------|
| 1 | 110°56.0910′ E/16°42.3414′ N | 580 m | No landing |
| 2 | 113°42.3066′ E/17°26.9924′ N | 100 m | No landing |
| 3 | 115°04.2688′ E/18°17.6657′ N | 1000 m | No landing |
| 4 | 115°04.2688′ E/18°17.6657′ N | 1800 m | No landing |
| 5 | 115°04.2688′ E/18°17.6657′ N | 2490 m | Walking on 30°slope |
| 6 | 115°06.1626′ E/18°17.7259′ N | 2493 m | Cutting and walking |
| 7 | 115°06.1626′ E/18°17.7289′ N | 2493 m | Cutting and walking |

Figure 15 shows the trajectories of the mining vehicle and support vessel during the tests.

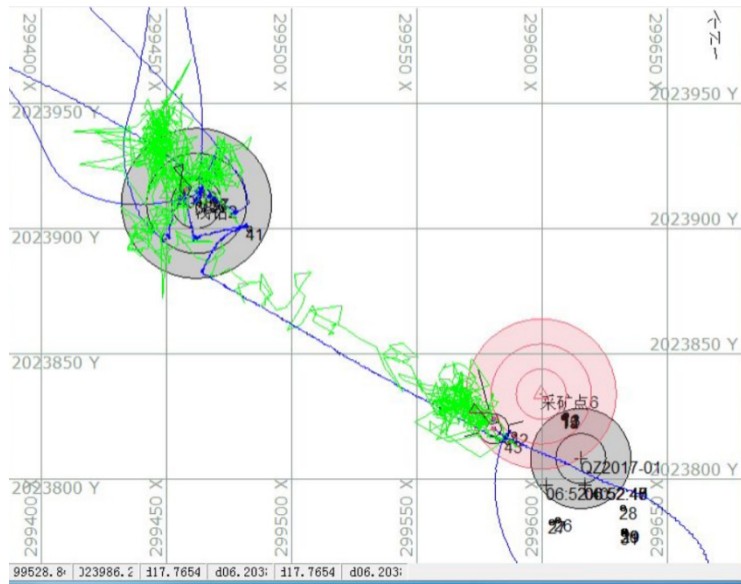

**Figure 15.** Trajectory diagram of the mining vehicle and support vessel (the green line is the vehicle's trajectory, and the blue line is the support vessel's trajectory).

## 6. Results

The results of the sea trial are summarized below:

(1) Walking function

The vehicle walk tests were conducted on the hard floor and soft floor, as shown in Figure 16a,b.

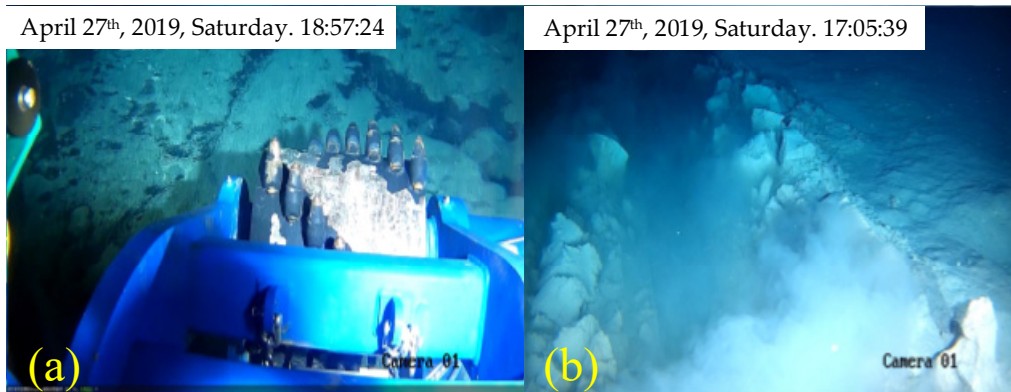

**Figure 16.** Walking function test: (**a**) hard floor, (**b**) soft floor.

The walking function performed well in these sea trials with total travel distance around 200 m and average speed of 0.03 m/s, including forward, backward and turning around.

(2) Crushing and Collection function

The crushing and collection were tested together in the sea trial. The crushing and collection system worked well and 6 kg of rocks were collected. Figure 17a shows that small particles, around 2 mm in diameter and 3 kg, were cut and collected by the vehicle in Test 6. Figure 17b is big particles, around 10 cm in diameter, 3 kg were cut and collected in Test 7.

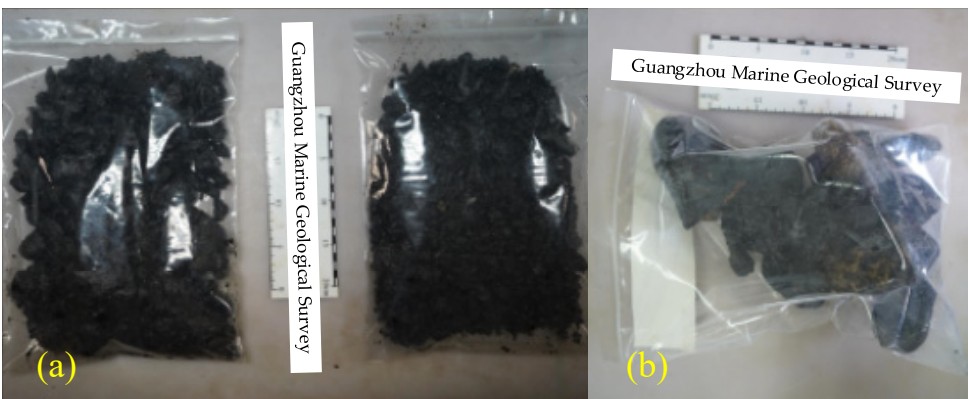

**Figure 17.** Particles: (**a**) small particles, (**b**) big particles.

(3) Awareness system and positioning and navigation system

Figure 18 shows the camera image during cutting tests. The plumes covered the sight of cameras as the actions of crushing and hydraulic collection. In this situation, the front image sonar was used to keep the vehicle away from big rocks. The positioning and navigation system worked well in the tests and plays an essential role in the launch and recovery process.

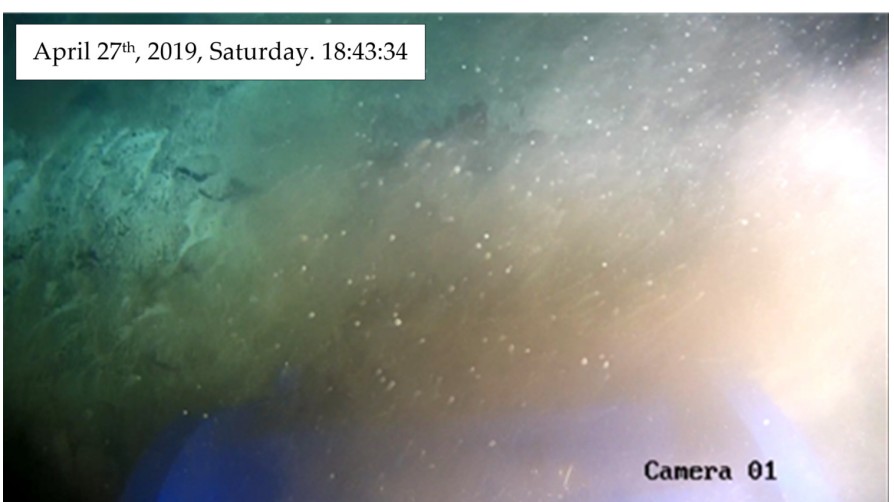

**Figure 18.** Camera image during cutting.

(4) HFSSP testing results

Figure 19 shows the sub-bottom profile results. The results of the ping from 1600 to 2100 are plotted in Figure 19a, while the top-right image, Figure 19b, is the single echo pulse compression curve at the ping 1720, in the green line, of Figure 19a. It can be seen from the curve that four strong scattering points are visible (red dots similar to the bottom image), which indicates the pulse compression performed well. The location of the surveying region was a mixture of sediments and rocks. The bottom figure shows a few segments with the stratum in the survey line represented by the green line ping with the hard substrate. Nonetheless, the tests data partially verified that HFSBP can measure a high-resolution seabed layer texture.

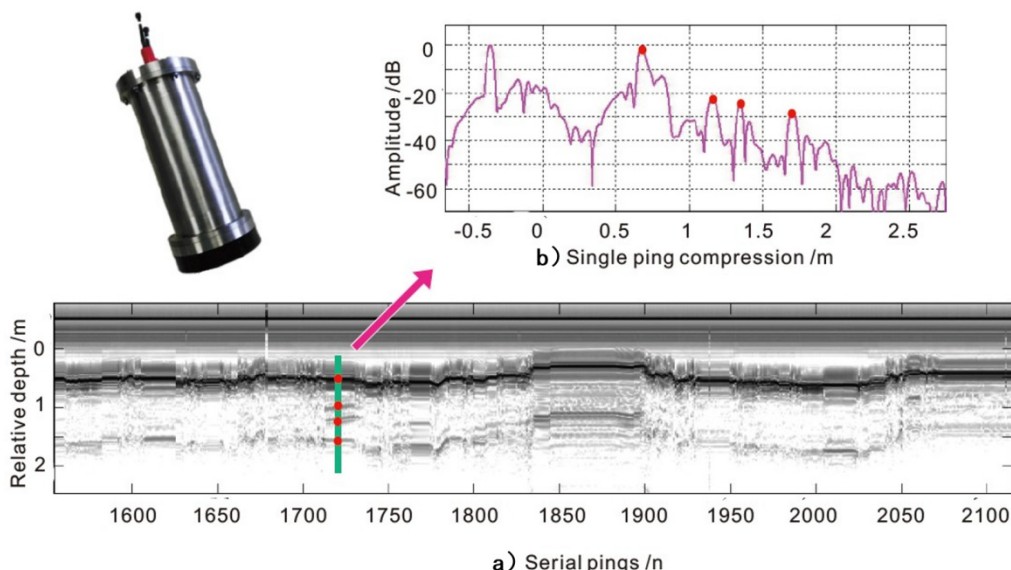

**Figure 19.** (**a**) HFSSP data in survey line and with (**b**) single ping compression.

## 7. Conclusions

This paper proposed a compact design of the subsea cobalt-rich crust mining vehicle with general purpose support vessel for subsea resource exploration, sample collection and research. The necessary functions were considered in the concept design, including walk, crushing/mining, sample collection, cutter head adaptation, vehicle orientation, crust texture measurement, awareness, positioning, and navigation. The prototype was tested in both tank and subsea environment. The following conclusions can be drawn:

(1)  The walk function worked well in both tank tests and sea trials. The track design worked well in the subsea environment and can be used for the subsea mining vehicle;

(2)  The crushing and collection functions worked well in sea trials. The crust was crushed to the size of 2 mm and 10 mm in Test 6 and 7, respectively. The sample size can be controlled by the setting of cutting parameters. The crushed crust was successfully collected by the jet pump up to 6 kg in total;

(3)  The crust texture was measured by the onboard sonar successfully, and clearly showed the texture from −0.5 m to 2.5 m. The onboard sonar can be used for the crust texture measurement;

(4)  The cameras captured the images of subsea environment, but the actions of crushing and sample collection produced plumes, which blocked the camera vision. In this situation, the front image sonar can be used to keep the vehicle away from big rocks;

(5)  The thrusters worked well in the launch and recovery processes. The thruster design can be used for the mining vehicle orientation.

Further works for this mining vehicle include the mechanical analysis and stability on slope (Part B); cutting parameters and ability of the cutter head (Part C); force-based cutter head adaptive control (Part D); jet pump CFD simulation and collection performance (Part E), etc.

**Author Contributions:** Conceptualization, N.Y. and C.X.; methodology, N.Y.; software, M.C. and J.Z.; validation, M.C., C.X., J.Z., C.A. and J.L.; formal analysis, C.X.; investigation, C.X; resources, Z.C.; data curation, Z.Q.; writing—original draft preparation, C.X.; writing—review and editing, S.Y., L.W. (Lan Wang) and L.W. (Liquan Wang); visualization, J.Z.; supervision, L.X.; project administration, C.A.; funding acquisition, Y.C. and L.X. All authors have read and agreed to the published version of the manuscript.

**Funding:** This research received no external funding.

**Acknowledgments:** This research was funded by China Merchants Deep Sea Equipment Research Institute (Sanya) Co., Ltd., Sanya, China. The authors gratefully acknowledge the effort made by the team of IDSSE, CAS, and the GMGS's sea trial support, and gratefully acknowledge the data analyzed by Xinghui Cao. Useful suggestions were given by Shaowei Zhang of IDSSE also acknowledged.

**Conflicts of Interest:** The authors declare no conflict of interest.

## Abbreviations

| | |
|---|---|
| ¡VAMOS! | Viable Alternative Mine Operating System |
| ROV | Remote Operated Vehicle |
| SMD | Soil Machine Dynamics Ltd. |
| JOGMEC | Japan Oil, Gas and Metals National Corporation |
| SMS | Seafloor Massive Sulfide |
| IDSSE | Institute of Deep-sea Science and Engineering |
| CAS | Chinese Academy of Sciences |
| GMGS | Guangzhou Marine Geological Survey |
| DPS | Dynamic Positioning System |
| SPS | Ship Positioning System |
| MBS | Multi-beam Bathymetry System |
| USBL | Ultra-Short Baseline |
| TOML | Tonga Offshore Mining Limited |
| HFSSP | High-Frequency Submersible Sub-Bottom Profiler |
| EVs | Electric Vehicles |
| NORI | Nauru Ocean Resources, Inc |
| CCZ | Clarion-Clipperton Zone |
| HMI | Human-Machine Interface |
| DSMF | Deep Sea Mining Finance Limited |

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
