# Peer review of "A Compact Design of Underwater Mining Vehicle for the Cobalt-Rich Crust with General Support Vessel Part A: Prototype and Tests"

_jmse, doi:10.3390/jmse10020135_

Round 1

Reviewer 1 Report

The topic of this paper is of interest to the readers of Journal of Marine Science and Engineering and is appropriate to the journal. Overall, this paper is well structured and the topic could be interesting but in general, the paper is not well written. I feel that the paper has major issues in order to be published. The comments and suggestions are given below:

1/ I am confident that the topic of the article is interesting and essential i terms of research. It should be noted however, the major concern is the novelty of the work. It is rather an implementation report than a scientific research paper. This work is contributive to engineering but does not bring insights in theory. Only a structural framework for engineering is provided, but no theoretical framework. Besides, a few novelties of the proposed algorithm are found in your description. This paper should have more theoretical contributions. 

2/ There are errors when authors mentioned the Figures in the text of this paper. The reviewer cannot realize which figures are described. Please check the entire paper again.

3/ The deep-sea mining vehicle usually consists of the tracked vehicle part and mining tool part, in the introduction part, the present research status of the mining tools is not sufficiently introduced. Recent applications of deep-sea mining vehicles should be introduced. For the mining tools (the chain wheel and ladder trencher), the authors can refer to the following papers: “Analytical design of an underwater construction robot on the slope with an up-cutting mode operation of a cutter bar”, “A study on an underwater tracked vehicle with a ladder trencher”, “A study on an up-milling rock crushing tool operation of an underwater tracked vehicle”, “Study on down-cutting ladder trencher of an underwater construction robot for seabed application”, “Analytical design of an underwater construction robot on the slope with an up-cutting mode operation of a cutter bar”, “Study on the Combined Underwater Tracked Vehicle System with a Rock Crushing Tool”, “Navigation and Control of Underwater Tracked Vehicle Using Ultrashort Baseline and Ring Laser Gyro Sensors”. Moreover, it is suggested that the novel index of this paper should be explained in detail. And the introduction should be added to do a better job of explaining the existing methods and why they are or are not valuable. It is important to make clear the new contribution of this manuscript to the related research field. Also, the authors should explain the situation of the market on underwater trencher systems and the difference of the proposed system with the commercialized products. 

4/ Why did the authors choose to study the specific mining vehicle proposed in this paper rather than other types of mining vehicles? Do the results in this paper can be used for other types of mining vehicles?

5/ Write the organization of the paper in the introduction part.

6/  All abbreviations need to be explained before using them in the manuscript.

7/ The underwater vehicle usually buries deep-sea mining vehicles while it works. Therefore, the effect of underwater cable is important to make its locomotion model. The authors can refer to the effects of cable on the motion of vehicles as: https://doi.org/10.3390/s20051329, “A study on hovering motion of the underwater vehicle with umbilical cable”. The authors should explain why its effect was neglected in the locomotion model of the authors. 

8/ Detailed implementation information should be provided (hardware, software, configuration, settings). A detailed discussion of hardware and software applied to the mining vehicle should be mentioned. Provide specifications of the hardware and software used for simulation of the approach. Because there is not enough data on this paper, the research results on the core idea of this paper seem unreliable.

9/  Very interesting illustrative material in the form of charts is not fully used because the text lacks chart interpretation. Extending this element will significantly affect the quality of this paper.

10/ The optimization problem should be better highlighted.

11/ The study of the proposed control system is missing in the paper. This needs to be incorporated.

12/ The underwater technique proposal is very weak, with some aspects not clearly specified such as dynamic model, control, path planning (such as the equations, the algorithms). The contribution of the positioning and navigation to underwater mining vehicles is minor. The theory of the method applied in the paper should be described carefully.

13/ Please improve the figure quality including font type, font, and style of variables. In Figure 9. Labels for each exponent graph should be displayed. The color contrast and quality of the images should be improved. The Label of the vertical and horizontal axis should be specified.

14/  In section 3.5 “Cost control”, please detail in more depth the accuracy of the numerical techniques for predicting the cost of the system. I suggest the authors give all the necessary manipulations for this. 

15/ The testing results are insufficient, the walking function test has no result in the paper, the author is required to elaborate on the analysis of the results. Besides, more results need to be added in the paper, such as the results of the heading angle of vehicle with respect to the time and position of the vehicle in forward, backward, and turning motions, especially, there are no results on the interaction of a rock crushing machine with the rock and the resulting forces and torques applied to the carriage of the machine. Also, the explanations and analysis of simulation results should be enriched to show the validity of the data.

16/ Results are presented without any quantitative and valuable discussion nor highlighting the limitation of the study. What are the assumptions? What are the problems to solve for a real-world implementation? The authors are able to clarify the few method's limitations, add a helpful conceptual model, and revise the discussion to clarify the interpretation and application of their results. 

17/ In Sections 4 and 5, rather than just displaying the experimental data, analysis of the testing results and comparisons with other state-of-the-art technologies will make the performance test more convincing. The reviewer would like to suggest the authors comparing, if possible, their results with some recently published work and clearly show the new design features in the current work.

18/ In section 6, the final paragraph should be re-written, please check again sentences 428-430.

19/  The Conclusion section is superficial, should include quantitative results, advantages, and disadvantages, limitation,s and recommendations for new implementations and future work.

20/  References should be edited according to the instructions for authors. Some author names are written in Italics. The reference format should be check-in this manuscript. 

21/ The English of the paper should be double-checked and revised since there are some mistakes in grammar and readability. Furthermore, the paper is not well written. Lots of formatting, grammatical issues.  I think that the English should be improved. There are many unclear sentences along the paper.

Reviewer 2 Report

- The article's main concept(s)

This paper presents the development of a deep-sea mining system. The authors design and implement a solution for cobalt rich crust subsea mining.

Raw materials are the new gold mines, all new technologies devices need raw materials, therefore deep-sea mining has lower impact on the planet and are being tested some exploitation sites.

The authors presented a solution that involved a surface vessel, an umbilical cable and the seafloor mining machine. This system was tested on a water tank and finally at 1000m and 2500m depth at sea trials.

The mining machine incorporates several hydraulic systems to deal with the mining process, and the awareness and navigation systems that uses imaging sonars, cameras, depth sensors, USBL, and an HFSSP system.

- Overall Comment

In overall,

 This document shows the importance of monitoring and impact assessment of deep-sea mining activities– and its key factors. The work has a good theoretical base, using useful references and information of general knowledge.

It presents a new design and all the developments on the Chinese raw material deep-sea mining program. The development of a prototype and low-cost mining machine.

The manuscript has a simple structure, very easy to read but some images are poor quality.

The conclusions resume the paper contribution and explain common sense knowledge.

There are several errors regarding the links to references.

- Weak and Strong points

Strengths

  • Great engineering development;
  • Deep-sea mining process – robotic platform;
  • A good resume about the new mining system design;
  • Marine science using new technologies;
  • ;

Weakness

  • The G forces are not explained were;

Round 2

Reviewer 1 Report

Thank you for the revised manuscript. In a general way, most of my comments were answered by the authors. Paper is done in a clear, understandable manner now. The manuscript now is acceptable for publishing